# The Impact of COVID-19 on Anxious and Depressive Symptomatology in the Postpartum Period

**DOI:** 10.3390/ijerph19137833

**Published:** 2022-06-26

**Authors:** Daniela Pereira, Brigite Wildenberg, Andreia Gaspar, Carolina Cabaços, Nuno Madeira, António Macedo, Ana Telma Pereira

**Affiliations:** 1Institute of Psychological Medicine, Faculty of Medicine, University of Coimbra, Rua Larga, 3004-504 Coimbra, Portugal; brigitewildenberg@hotmail.com (B.W.); csm.cabacos@gmail.com (C.C.); nunogmadeira@gmail.com (N.M.); amacedo@ci.uc.pt (A.M.); apereira@fmed.uc.pt (A.T.P.); 2Department of Psychiatry, Centro Hospitalar e Universitário de Coimbra, Praceta Professor Mota Pinto, 3004-561 Coimbra, Portugal; 3Department of Gynecology and Obstetrics, Bissaya Barreto Maternity Hospital, Centro Hospitalar e Universitário de Coimbra, 3000-045 Coimbra, Portugal; andreiadevasconcelosgaspar@gmail.com

**Keywords:** COVID-19, perinatal depression, perinatal anxiety, fear of COVID-19

## Abstract

Background: Women in the postpartum period may be particularly vulnerable to the psychological effects of the COVID-19 pandemic. The aim of our study was to evaluate the impact of the coronavirus pandemic on postpartum depression and anxiety levels and the role of the fear of COVID-19 in its development. Methods: Women who delivered at the Bissaya Barreto Maternity Hospital, between 16 March and 16 June 2020 (Group 1: Birth in COVID-19 period, n = 207), recruited in the postpartum period, filled in a set of self-reported validated questionnaires: Perinatal Depression Screening Scale, Perinatal Anxiety Screening Scale, Profile of Mood States, Perseverative Thinking Questionnaire, Dysfunctional Beliefs Towards Maternity Scale, and the Fear of COVID-19 Scale. Levels of depressive and anxious symptomatology, negative affect, negative repetitive thinking, and the dysfunctional beliefs towards motherhood of these women were compared with data from samples of previous studies that included women whose delivery had occurred at the same Maternity Hospital before the COVID-19 pandemic period (Group 2: Birth before the COVID-19 period, n = 212). Results: Based on the cutoff points of the screening scales, the prevalence of clinically relevant depressive and anxious symptoms in Group 1 was 40.1% and 36.2%, respectively. Women in Group 1 had significantly higher levels of anxious and depressive symptoms, negative affect, negative repetitive thinking, and dysfunctional beliefs towards motherhood than women in Group 2 (*p* < 0.05). Fear of COVID-19 in the postpartum period was a predictor of depressive (ß = 0.262) and anxious (ß = 0.371) symptoms, explaining 6.9% and 13.7% of their variability, respectively (*p* < 0.001). Conclusion: During the COVID-19 pandemic, women in the postpartum period present greater depressive and anxious symptomatology, as well as increased risk factors.

## 1. Introduction

In late December 2019, several local health authorities of Wuhan, Hubei Province in China, reported clusters of patients with pneumonia of an unknown cause, which were linked to a seafood market in Wuhan [1]. The pathogen, a novel coronavirus (SARS-CoV-2), was identified by local hospitals, as stated by the WHO on 9 January 2020. Subsequently, COVID-19 has spread rapidly throughout the world and has reached pandemic proportions, affecting all continents. The WHO declared the outbreak a public health emergency of international concern on 30 January 2020 [2]. On 11 March 2020, the outbreak was declared a global pandemic.

Severe disruption of normal life due to COVID-19, fear of contracting the disease, and anticipation of negative economic consequences led to increased symptoms of depression, anxiety, and stress in the general population [3], and women in the perinatal period were no exception. 

Even in normal circumstances, the perinatal period, ranging from pregnancy to one year postpartum, is characterized by physical, psychological, and relational changes, being marked by unpredictability [4] and often perceived as stressful [5]. These changes lead women to be more vulnerable than ever to the development of depressive and anxiety disorders [6]. 

Approximately 20–30% of women worldwide experience at least one psychiatric disorder during pregnancy or postpartum [7]. About 13–21% of prenatal and 11–17% postpartum women experience anxiety and depression [8,9].

The etiology of perinatal mental health problems is multifactorial. The main risk factors for perinatal depression and anxiety are previous history of depressive and or anxiety disorder; depression and or anxiety in pregnancy; negative affect, insomnia; lack of partner support during pregnancy; and the existence of stressful life event in the previous year [6,9,10,11].

During the COVID-19 pandemic, women were in an even more vulnerable position due to the preventive measures that were implemented, such as the following: the constrained authorization of the companion’s presence during childbirth (depending on the institution’s guarantee of safety conditions); women were advised to remain in an isolated area in the postpartum and to only be accompanied by designated health professionals; due to the risk of contagion to the newborn, skin-to-skin contact between mother and baby was restricted to mothers who opted to do so after careful consideration of the potential risks [12]. Additionally, new mothers’ concerns about their own and their baby’s health, risk of possible transmission of infection to the baby, changes in babies’ healthcare (e.g., changes in appointments and restrictions of companions), reduced social support (as visits were discouraged to limit virus propagation), and assistance (due to travel restrictions) were important additional stress sources. Moreover, many pregnant and postpartum women and their partners were also dealing with additional tasks such as childcare and home schooling, if there were other children at home, because schools and children centers had closed in many places, or families chose to keep them at home due to fear of contamination [13].

Taking these factors into account, it is reasonable to assume that the highly stressful experience of the COVID-19 pandemic could increase the risk of a perinatal mental health problem.

Some published studies analyzed the impact of the COVID-19 pandemic on depressive and anxiety symptoms during the perinatal period, as well as contributing risk factors for depression and anxiety in this period, with some contradictory results.

A recent systematic review with meta-analysis evaluated the effect of the COVID-19 pandemic on the anxiety and depression of women during pregnancy and the perinatal period, providing evidence that the COVID-19 pandemic significantly increases the risk of anxiety among women during pregnancy and the perinatal period [14]. 

A study by Perzow et al. (2021) that longitudinally examined changes in internalizing symptoms from before to during the COVID-19 pandemic among pregnant and postpartum women concluded that depressive symptoms were higher during COVID-19 compared to pre-COVID-19 and were just as high as during early pregnancy; concerning anxiety symptoms, these were higher during COVID-19 compared to both pre-COVID-19 and early pregnancy. This study also investigated moderation by loneliness and other contextual risk factors, concluding that higher loneliness was associated with increased depressive symptoms during COVID-19, and lower income-to-needs-ratio most strongly predicted symptoms during early pregnancy [15]. 

Oswin et al. (2021) compared levels of depression and anxiety, the quality of social relationships, and the temperament of infants of treatment-seeking mothers for postpartum depression prior to and during the pandemic, noting that the pandemic may not have worsened depression, anxiety, social relationships, or relationships with partners, but appears to have contributed to poorer mother–infant interactions and maternal reports of more negative emotionality in their infants [16]. 

Literature on postpartum women as well as contributing risk factors for postpartum depression and anxiety, especially considering the role of the fear of COVID-19 in the pandemic period, remains scarce.

According to Pakpour and Griffiths (2020), without knowing the level of fear about COVID-19 among different groups by specific socio-demographic variables and/or different psychological factors, it is difficult to know whether education and prevention programs are needed, which groups to target, and where [17]. Additionally, it is critical to understand how this fear impacts mental health to design preventive and selected programs. 

The aims of this study are to assess the impact of the COVID-19 pandemic on the levels of postpartum depression and anxiety, as well as to possibly correlate and risk factors and to analyze the role of fear of COVID-19 as a potential predictor of this symptomatology.

If the study confirms our hypothesis that women whose delivery occurred during the COVID-19 pandemic have higher levels of depressive and anxiety symptoms and more fear of COVID-19, as well as of other known psychological risk factors for them, it can be a relevant contribution for the development of preventive strategies directed at the identified risk factors throughout the current or future pandemics.

## 2. Material and Methods

This study is part of an ongoing research project approved by the Ethical Committee of the Centro Hospitalar e Universitário de Coimbra (CHUC) (Reference: CHUC-166-20).

### 2.1. Procedure and Participants

This cross-sectional and correlational study was conducted at Bissaya Barreto Maternity Hospital (BBMH) of the Centro Hospitalar e Universitário de Coimbra, in Portugal.

Eligible participants included in the study were women over 18 years old, with good command of spoken and written Portuguese, who had given birth between 16 March and 16 June 2020. Exclusion criteria were reported complications during pregnancy and/or during labor and the occurrence of fetal death. Of the 441 contacted women, 207 postpartum women were included in Group 1 (women who gave birth during the first wave of COVID-19 at the BBMH in Coimbra, Portugal). The total number of applicants and reasons for exclusion are summarized in Figure 1.

Recruitment was conducted by telephone, between January and March 2021, and those who agreed to participate received an email with a link to a Google Form Survey, which contained questions related to sociodemographic and psychosocial variables, COVID-19 related questions, and validated self-report questionnaires: the Perinatal Depression Screening Scale, the Perinatal Anxiety Screening Scale; Profile of Mood States; Perseverative Thinking Questionnaire, the Dysfunctional Beliefs Towards Motherhood Scale; and the Fear of COVID-19 Scale.

Group 2 was composed of 212 women who had participated in previous studies from our group. These women were recruited during pregnancy, between 2018 and 2019, and had also delivered at BBMH, outside the pandemic period. They were evaluated during pregnancy, and in three moments after delivery (2, 6, and 12 months), and had filled out online the same set of sociodemographic and psychosocial related questions and validated questionnaires of the women from Group 1, except for the Fear of COVID-19 Scale. Inclusion and exclusion criteria were the same as for those of Group 1. We chose to include in Group 2 data obtained from the 6-month postpartum evaluation because they were more approximate to the evaluation time of the Group 1 women.

The applicants were informed about the procedure and aims of the study, and the confidentiality and anonymity of collected data was guaranteed. Online informed consent was obtained by asking all participants to click a button within the online survey to consent to participate.

### 2.2. Measures

All the questionnaires revealed good reliability and validity (construct and concurrent) in samples of Portuguese women in the postpartum period. 

#### 2.2.1. Depressive Symptoms 

The Portuguese short version of the Postpartum Depression Screening Scale (PDSS; Beck and Gable, 2000 [18]; Portuguese validation by Pereira et al., 2013 [19]) was used to assess depressive symptoms in the previous month. This scale consists of twenty-one items with responses ranging from 1 (disagree a lot) to 5 (I very much agree). According to the validation study, a cutoff value of 43 was used in the current study to define postpartum depressive symptoms (PDS) [19]. 

Cronbach’s α for the PDSS-21 was 0.952 for Group 1 and 0.940 for Group 2.

#### 2.2.2. Anxiety Symptoms 

The Perinatal Anxiety Screening Scale (PASS; Somerville et al., 2014 [20]; validated for Portuguese women in the postpartum period by Pereira et al., 2020 [21]) was used to assess the severity of anxiety symptoms in the previous month. This scale consists of thirty-one items, with responses ranging from 0 (never) to 3 (almost always). According to the validation study, a cutoff value of 30 was used in the current study to define postpartum anxious symptoms (PAS) [21].

Cronbach’s α for the PASS-31 was 0.960 for Group 1 and 0.950 for Group 2.

#### 2.2.3. Negative Affect 

The Portuguese short versions of Profile of Mood States (POMS; McNair et al., 1971 [22]) is commonly used as a measure of perinatal psychological distress [23]. The version of the scale used to evaluate Negative Affect (NA) and Positive Affect (PA) in this study consist of eighteen items with responses ranging from 0 (not at all) to 4 (extremely) [23].

Cronbach’s α for the POMS-18 was 0.866 for Group 1 and 0.833 for Group 2.

#### 2.2.4. Negative Repetitive Thinking

The sum of the 15 items of the Portuguese version [24] of the Perseverative Thinking Questionnaire (PTQ; Ehring et al., 2011 [25]) was used as a global score of repetitive negative thinking (RNT).

Cronbach’s α for the PTQ-15 was 0.978 for Group 1 and 0.975 for Group 2.

#### 2.2.5. Dysfunctional Beliefs towards Motherhood

The Attitudes Toward Motherhood Scale (AToM; Warner et al., 1997 [26]; validated for Portuguese postpartum women by Costa et al., 2017) was used to assess Dysfunctional Beliefs Towards Motherhood (DBTM) [27]. This scale comprises three subscales, according to the target of the participants’ beliefs: Beliefs related to the judgment of others (BRJO); Beliefs related to the idealization of maternal role (BRIMR); and Beliefs related to maternal responsibility (BRMR). The Likert scale, of 12 items, ranges from 1 (always disagree) to 6 (always agree).

Cronbach’s α for the AToM was 0.888 for Group 1 and 0.885 for Group 2.

#### 2.2.6. Fear of COVID-19

The Fear of COVID-19 Scale (FCV-19S; Ahorsu et al., 2020 [28]) is a self-report measure aimed at assessing fear of COVID-19. As the original, the Portuguese version for postpartum women (FCV-19PS) consists of seven items assessing emotional fear reactions towards the pandemic (e.g., “I am most afraid of Corona”) and presented good construct validity and internal consistency as a unidimensional measure [29]. For the purpose of this study, we have added another item, specifically related to the postpartum context (8. “I’m afraid that my baby will be infected with COVID-19”). This adapted version was also revealed to be unidimensional and presented good construct and concurrent validity [30].

Cronbach’s α for the FCV-19PS was 0.880.

#### 2.2.7. Other Variables

Other variables assessed in this study included sociodemographic factors (maternal age, nationality, marital status, educational qualification, employment status); obstetric factors (parity, mode of delivery, complications during pregnancy and/or during labor, neonatal birth weight); mental health history (personal history of depression and/or anxiety). Personal mental health history was assessed by a single item with a dichotomous response (“yes” vs. “no”): “have you ever experienced a depressive or anxious episode?”

Other variables included were COVID-19-related factors, such as incidence of COVID-19 in the mother and any family members, and a dichotomic question (“yes” vs. “no”) related to maternal fear of COVID-19 during pregnancy (e.g., “During pregnancy were you afraid that something would happen to your baby in pregnancy, childbirth, or postpartum due to COVID-19?”). Marital relationship and perception of support from the partner and other family members were also assessed.

### 2.3. Statistical Analyses

All statistical analyses were performed using IBM Statistical Package for the Social Sciences (SPSS Version 26 for Mac). First, Cronbach’s alpha coefficients were calculated to assess the internal consistency of all the measures used. Descriptive analyses for demographic, clinical, psychosocial, and COVID-19 specific measures were conducted. Independent two-sample *t*-tests and chi-squared tests for categorical variables were used to compare the two groups. 

Descriptive analyses were also used to identify the prevalence rates of clinically relevant cases of depression and anxiety in both groups, according, respectively, to PDSS [19] and PASS validated cutoff scores [21].

Pearson’s correlation coefficients were used to test the association between continuous variables, and Spearman’s correlation coefficients were performed when one or more variable was dichotomous/categorical.

Hierarchic linear regression analyses were conducted to examine if and how much the Fear of COVID-19 explained the levels of the depressive and anxious symptoms reported by Group 1. Statistical significance was set at a *p*-value of <0.05.

## 3. Results

### 3.1. Sample Characteristics

Group 1 (Birth in COVID-19 pandemic period) was composed of 207 women with a mean age of 33.51 ± 5.23 years. 

Almost all women (98%) considered themselves as having a reasonable or good relationship with their partner, and 96.1% felt they had partner support. Regarding family support, 88.4% believed they had support.

Group 2 (Birth outside COVID-19 pandemic period) was composed of 212 women with a mean age of 32.21 ± 4.52 years. 

The majority of women (96.7%) considered themselves as having a reasonable or good relationship with their partner, and 92.2% felt they had partner support. Regarding family support, 96.7% believed they had support.

Group 1 and Group 2 were evaluated at significantly different times after delivery (8.59 months ± 0.94 vs. 6.72 months ± 1.36; *p* < 0.001). The two groups did not differ regarding their sociodemographic, psychosocial, and clinical features, except for their perception of family support. These variables are summarized in Table 1.

### 3.2. COVID-19 Related Features

Regarding incidence of COVID-19 in the mother or family members of women in Group 1, 14 women had a history of previous or current COVID-19 infection and 26 had an infected family member.

One hundred seventy women (82.1%) reported having been afraid during pregnancy that something would happen to their baby in pregnancy, childbirth, or postpartum due to COVID-19.

The mean score of FCV-19S among the COVID-19 study group was 22.72 ± 7.07.

### 3.3. Depressive and Anxious Symptoms and COVID-19

Of the women from Group 1, 40.1% scored above the cutoff point of 43 in the PDSS, and 36.2% had scores above the cutoff point of 30 in the PASS, as opposed to women in Group 2, where only 9.8% and 8.5% scored above the cutoff points in the PDSS and PASS (χ^2^= 20.32; χ^2^ = 17.61, *p* < 0.001), respectively.

Several *t*-tests were run to explore the effects of COVID-19-related features on the mean levels of depressive and anxious symptomatology in Group 1 women. Postpartum depressive symptoms (PDS) were significantly higher in women who reported having been afraid during pregnancy that something would happen to their baby in pregnancy, childbirth, or postpartum due to COVID-19 (43.66 ± 15.52 vs. 36.00 ± 14.14; *p* = 0.028). The women’s own or family’s history of infection had no effect on PDS, nor on postpartum anxious symptoms (PAS).

Correlational analyses were performed for continuous variables (see Table 2 and Table 3 for Pearson correlation values and significance). Results showed a significant positive correlation between PDSS and PASS scores, negative affect (NA), negative repetitive thinking (NRT), dysfunctional beliefs towards motherhood (DBTM), fear of COVID-19 in the postpartum period (FCV-19P), and marital relationship (MR). Statistically significant negative correlations were found between PDSS and perceptions of partner’s support (PS) and family support (FS).

As for PASS, a significant positive correlation between its scores and scores of PDSS, NA, NRT, DBTM, FCV-19P, and MR was found.

Group comparison analysis showed that PDSS mean scores were higher in Group 1 than in Group 2 (42.29 ± 17.19 vs. 34.01 ±13.19; *p*< 0.001). The same was observed for PASS (26.96 ± 18.29 vs. 18.50 ± 13.19; *p* < 0.001). Women in Group 1, when compared to Group 2, had significantly higher levels of NA, NRT, and DBTM. See Table 4 for means and standard deviations.

Lastly, to examine if and how much the fear of COVID-19 explained the levels of PDS and PAS reported by Group 1, a hierarchic linear regression was performed. Because perceived positive family support was significantly lower in Group 1 (Table 1), and this is a known risk factor for both perinatal anxiety and depression [11], it was entered in the first step of the model. After controlling for this psychosocial variable, fear of COVID-19 still accounted for a significant increment in the PDS and PAS variance, respectively, of 6.8% and 13.7% (*p*< 0.001) (Table 5).

## 4. Discussion

This study investigated the impact of the COVID-19 pandemic on depressive and anxious symptoms of Portuguese mothers who had given birth during the first wave of the pandemic. One of the most relevant findings was that, based on the cutoff points of the screening scales, the prevalence of clinically significant symptoms of depression and anxiety was higher in these women compared to those that had given birth before the pandemic period, suggesting that postpartum women during the COVID-19 pandemic period are at greater risk of developing postpartum depression and anxiety.

Higher proportions of depressive and anxious symptoms in the postpartum period have been reported in other studies on the same topic [31,32,33,34], even though those studies had used different screening scales, such as the Edinburgh Postnatal Depression Scale (EPDS) [31,32,34] and the State-Trait Anxiety Inventory (STAI) [31,32,33].

Several COVID-19-related variables have been shown to have a significant effect on PDSS and PASS scores. More specifically, pregnant women who reported being afraid that something bad would happen to their baby during pregnancy, childbirth, or postpartum due to COVID-19 had significantly higher postpartum depressive symptoms. Additionally, being afraid of COVID-19 in the postpartum period significantly correlated with depressive and anxious symptoms, as well as with levels of negative affect, negative repetitive thinking, and dysfunctional beliefs towards motherhood. Fear of COVID-19 explains a significant variability of depressive and anxious symptoms.

These results are in line with previous studies that showed that intense/persistent fear may lead to the development of common mental disorders, such as depression, anxiety, and substance use disorders [35].

To the best of our knowledge, only one other study has evaluated the impact of fear of COVID-19 on depressive symptoms of postpartum women [36], assessing features such as loss of job, perceived support from family, infection of themselves, infection of close others, fear of being infected, fear of a close one being infected, and fear of the child being infected. The results of that study pointed out a statistically significant effect of having been infected by the virus, having had a close one infected or having been in contact with infected ones, and fear of their child being infected on the total scores of the EPDS.

In our study, the women’s own infection or infection of relatives were not related with fear of COVID-19 in the postpartum period, neither with depressive nor anxious symptoms. We can only speculate that what worries the recent mothers the most is their baby’s safety, and that the fear that something will happen to their babies outweighs the fear of what might happen to others. We believe that more studies are needed to better understand the mechanisms underlying the fear of COVID-19 in this population.

Besides COVID-19′s related factors, other previously identified risk factors for postpartum depressive symptoms, such as negative affect, negative repetitive thinking, and dysfunctional beliefs towards motherhood, significantly correlated with depressive and anxious symptoms and were significantly higher in postpartum women of the COVID-19 pandemic period, further emphasizing the greater risk for developing postpartum depression and anxiety in these women [37,38].

As for marital relationships and perception of support, according to our results women who gave birth during the COVID-19 pandemic differed from those who had given birth outside the pandemic period in terms of the perception of family support, with the first ones feeling less supported. This is not surprising, as during the pandemic social support was severely limited due to the restrictions that had been put into place to reduce the transmission of COVID-19 in hospital settings [39], which had an impact on PDS. Women who felt less supported by their partners also tended to have more PDS. Surprisingly, women who reported a good marital relationship had significantly more depressive and anxious symptomatology.

It came as no surprise that a stressful event in the last year enhances the risk of perinatal mental health problems, especially postpartum depression, as this relationship has been widely studied [10,40,41]. However, the impact of the COVID-19 pandemic on postpartum women’ s mental health has not yet been fully explored. Our study identified some factors that put this population at greater risk for the development of postpartum depression and postpartum anxiety, and to the best of our knowledge this is the first study to evaluate fear of COVID-19 in postpartum women using a validated questionnaire and the role of this fear in postpartum depressive and anxious symptoms. 

Hopefully, it will contribute to the understanding of how devastating the impact of pandemics can be in the mental health of women in the perinatal period.

During the pandemic period, preventive and therapeutic measures that avert the spread of the virus should be prioritized. Cognitive behavioral therapy (CBT) is recommended for the treatment of mild to moderate anxiety and/or depression in postpartum women. Delivering CBT via the Internet (iCBT) could be a way to overcome barriers to treatment access and therefore to improve treatment coverage. Additionally, Brief unguided iCBT programs that can be accessed without relying on mental health clinicians for instruction may offer a more scalable, cost-effective way to teach new mothers how to manage anxiety and depression. A brief, unguided iCBT intervention, ‘MUMentum Postnatal’, developed to target symptoms of anxiety and depression in postpartum women, found that this postnatal program was highly effective, showing significantly greater reductions in anxiety, depression, and psychological distress when compared with usual care. Additionally, the program produced meaningful improvements in maternal bonding, confidence in parenting, and quality of life, with high participant engagement, adherence, and treatment satisfaction [42].

We acknowledge several limitations of this study. First, the assessments of depressive and anxious symptoms relied on self-reported measures, which may be associated with response bias, namely social desirability bias. However, the fact that the questionnaires were answered online and anonymously may have contributed to reducing that bias. Additionally, despite good specificity and sensibility for identifying depressive and anxious symptoms in the perinatal period [19,21], PDSS and PASS do not provide a diagnosis of postpartum depression or postpartum anxiety disorder, respectively. Furthermore, due to the cross-sectional design of the study, no definite conclusions can be drawn concerning the direction of causality. Finally, the sample size is small and geographically specific, and thus the results cannot be generalized.

The significant difference in the mean evaluation time (measured in months after delivery) can be pointed out as a potential limitation, as depressive symptoms, mainly somatic-affective, seem to decrease as women adapt to hormonal and behavioral changes associated with pregnancy and childbirth [43]. However, previous research by our group showed that cognitive-affective symptoms, measured with the PDSS, did not differ between the sixth and twelfth months postpartum in women, both with and without depression [43].

Additionally, it is important to take in account that possible confounding factors such as individual factors that can influence the fear of COVID-19, namely the perception of fear and some personality traits (e.g., neuroticism and perfectionism), [44] were not considered. Additionally, other COVID-19-related factors in the perinatal period, such as vaccination, were not considered. A recent study observed that the most common reason for pregnant women to refuse vaccination was “Worry about the safety of the vaccine” [45]. That uncertainty might contribute to increased anxiety in pregnant and postpartum women.

Nevertheless, the findings from the present study are relevant and should not be invalidated, especially because the general demographic variables were similar between both groups.

In the future, multi-center longitudinal studies, using clinical structured interviews to assess postpartum depression and anxiety and to enable generalization of results, should be considered. Additionally, it would be noteworthy to explore potential mediator factors in the relationship between fear of COVID-19 and postpartum depressive and anxious symptoms, so that specific preventive strategies can be implemented.

## 5. Conclusions

During the COVID-19 pandemic, postpartum women presented higher levels of depressive and anxious symptoms, as well as of the risk factors for them. Anxiety and depression tend to be higher in women with greater levels of fear of COVID-19, negative affect, negative repetitive thinking, and dysfunction beliefs towards motherhood.

Because the fear of COVID-19 was found to explain the significant variability of postpartum depression and anxiety symptoms, postpartum women who show great fear should be further screened for these symptoms.

Our findings could be applied to identify patients at greater risk of experiencing adverse mental effects, and they also fully support the idea that preventive measures, screening, and access to specialized mental health services for mothers need to be intensified during pandemic periods.

## Figures and Tables

**Figure 1 ijerph-19-07833-f001:**
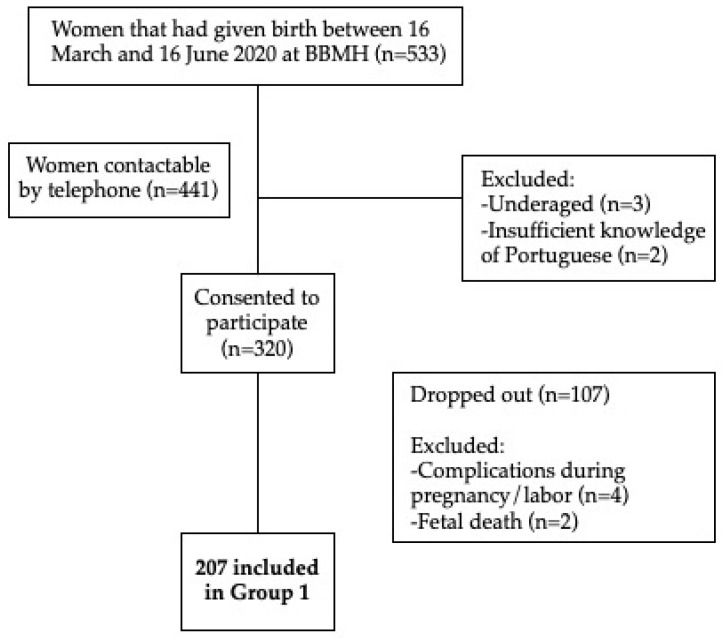
Flow chart of recruitment and screening.

**Table 1 ijerph-19-07833-t001:** Sociodemographic characteristics and psychosocial and clinical features of mothers who gave birth in COVID-19 period (Group 1) and mothers who gave birth outside COVID-19 period (Group 2).

Characteristics	Group 1: Birth in COVID-19 Period (n = 207)	Group 2: Birth Outside COVID-19 Oeriod (n = 212)	*p*-Value
Maternal age, Y	33.51 ± 5.23	33.21 ± 4.52	0.526
Nationality			
Portuguese	190 (91.7)	194 (91.5)	0.918
Level of education			0.520
Basic	8 (3.8)	15 (7.1)	
Secondary	61 (29.5)	56 (26.4)	
Higher	138 (66.7)	141 (66.5)	
Civil Status			0.173
Single	31 (14.9)	48 (22.6)	
Married/Cohabitating	173 (83.5)	157 (74.1)	
Divorced/separated	3 (1.4)	7 (3.3)	
Occupation			0.057
Employed	164 (79.2)	163 (76.9)	
Unemployed	28 (13.5)	20 (9.4)	
Maternity leave	15 (7.2)	29 (13.7)	
MR			0.200
Reasonable/good	203 (98)	205 (96.7)	
Bad	4 (2)	7 (3.3)	
PS			0.999
Positive	199 (96.1)	204 (92.2)	
Negative	8 (3.9)	8 (3.8)	
FS			0.001
Positive	183 (88.4)	205 (96.7)	
Negative	24 (11.6)	7 (3.3)	
Parity			0.069
Primiparous	117 (56.5)	101 (47.6)	
Mode of delivery			0.817
Eutocic	105 (50.7)	116 (54.7)	
Dystocic	49 (23.7)	40 (18.9)	
Cesarian	53 (25.6)	56 (26.4)	

**Table 2 ijerph-19-07833-t002:** Correlations of the Postpartum Depression Screening Scale (PDSS) and the Postpartum Anxious Screening Scale (PASS) scores and Negative Affect (NA), Negative Repetitive Thinking (NRT), Fear of COVID-19 in Postpartum period (FCV-19P), Dysfunctional Beliefs Towards Motherhood (DBTM), marital relationship (MR), and perception of partner’s support (PS) and family support (FS) for Group 1.

Variables	2.	3.	4.	5.	6.	7.	8.	9.
PDS	0.660 **	0.506 **	0.672 **	0.519 **	0.262 **	0.256 **	−0.199 **	−0.217 **
2. PAS	1	0.595 **	0.740 **	0.541 **	0.371 **	0.154 *	−0.177	−0.120
3. NA		1	0.520 **	0.319 **	0.262 **	0.222 **	−0.223 **	−0.143 *
4. NRT			1	0.581 **	0.316 **	0.173 *	−0.193 **	−0.157 *
5. DBTM				1	0.283 **	0.135	−0.146 *	−0.122
6. FCV-19P					1	−0.051	0.029	−0.008
7. MR						1	−0.401 **	−0.171 *
8. PS							1	0.493 *
9. FS								1

Abbreviations: PDS, Postpartum Depression Symptoms; PAS, Perinatal Anxiety Symptoms; NA, Negative Affect; NRT, Negative Repetitive Thinking; DBTM, Dysfunctional Beliefs Towards Motherhood; FCV-19P, Fear of COVID-19 in Postpartum; MR, marital relationship; PS, perception of partner’s support; FS, perception of family’s support. Note, * *p* < 0.05; ** *p* < 0.01.

**Table 3 ijerph-19-07833-t003:** Correlations of Postpartum Depression Screening Scale (PDSS) and Postpartum Anxious Screening Scale (PASS) scores and Negative Affect (NA), Negative Repetitive Thinking (NRT), Dysfunctional Beliefs Towards Motherhood (DBTM), marital relationship (MR), and perception of partner’s support (PS) and family support (FS) for Group 2.

Variables	2.	3.	4.	5.	6.	7.	8.
PDS	0.700 **	0.725 **	0.608 **	0.551 **	−0.033	−0.185 *	−0.295 **
2. PAS	1	0.628 **	0.774 **	0.616 **	−0.035	0.216 *	−0.193 *
3. NA		1	0.575 **	0.373 **	−0.268 **	0.285 *	−0.411 **
4. NRT			1	0.498 **	0.022	0.114	−0.184 *
5. DBTM				1	0.132	0.033	−0.139
6. MR					1	−0.019	0.003
7. PS						1	−0.498 **
8. FS							1

Abbreviations: PDS, Postpartum Depression Symptoms; PAS, Perinatal Anxiety Symptoms; NA, Negative Affect; NRT, Negative Repetitive Thinking; DBTM, Dysfunctional Beliefs Towards Motherhood; MR, marital relationship; PS, perception of partner’s support; FS, perception of family’s support. * *p* < 0.05; ** *p* < 0.01.

**Table 4 ijerph-19-07833-t004:** Comparison of depressive and anxiety symptomatology, negative affect, negative repetitive thinking, and dysfunctional beliefs towards motherhood scores between Group 1—Mothers who gave birth in COVID-19 period, and Group 2—Mothers who gave birth outside COVID-19 period—Students *t*-Test.

	Group 1: Birth in COVID-19 Period (n = 207)	Group 2: Birth Outside COVID-19 Period (n = 212)	*p*-Value
PDS	42.29 ± 17.19	34.01 ± 13.19	<0.001
PAS	26.96 ± 18.29	18.50 ± 13.68	<0.001
NA	14.60 ± 9.84	9.21 ± 9.48	<0.001
NRT	16.49 ± 14.54	11.84 ± 11.87	<0.001
DBTM	30.85 ± 12.61	25.62 ± 11.63	<0.001

Abbreviations: PDS, Postpartum Depression Symptoms; PAS, Perinatal Anxiety Symptoms; NA, Negative Affect; NRT, Negative Repetitive Thinking; DBTM, Dysfunctional Beliefs Towards Motherhood. Values are given as mean ± SD.

**Table 5 ijerph-19-07833-t005:** Hierarchic linear regression analyses of the fear of COVID-19 on depressive and anxious symptoms reported by Group 1.

DV	IV	R^2^	R^2^ Change	Beta	*p*-Value
PDS	FS	4.2%	----	−0.217	<0.001
	FCV-19P	10.6%	6.8%	0.260	<0.001
PAS	FS	1%	-----	−0.120	<0.001
	FCV-19P	14.3%	13.7%	0.370	<0.001

Abbreviations: DV, dependent variable; IV, independent variable; PDS, Postpartum Depression Symptoms; PAS, Perinatal Anxiety Symptoms; FCV-19P, Fear of COVID-19 in postpartum; FS, Family support.

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
