# Peer review of "The Impact of COVID-19 on Anxious and Depressive Symptomatology in the Postpartum Period"

_ijerph, 2022, doi:10.3390/ijerph19137833_

Round 1

Reviewer 1 Report

Thank you for the possibility to review the manuscript titled: “The impact of COVID-19 in anxious and depressive symptomatology in the postpartum period”. The study is interesting and easy to read. There are several minor corrections:

-Please review the language of the manuscript as it has several minor mistakes.

-Figure 1 is visible only partially.

P 170-171 the text is inconsistent, please check the blank spaces.

Number one indicated before every section such as results, discussions etc.

Please take into account the recommendations in the spirit of improving the quality of the submission

Author Response

Thank you very much for the pertinent and useful revision and for the opportunity of improving our paper entitled “The impact of COVID-19 in anxious and depressive symptomatology in the postpartum period”.  We have appreciated the comments and have accepted the suggestions.

Please find below our answers and the list of changes point by point.

-Please review the language of the manuscript as it has several minor mistakes.

R:  The language was revised through the manuscript (at blue in the manuscript).

-Figure 1 is visible only partially.

R: Figure 1 was adjusted.

P 170-171 the text is inconsistent, please check the blank spaces.

R: The blank spaces were checked

Number one indicated before every section such as results, discussions etc.

R: The numbering was corrected.

Best regards

Reviewer 2 Report

The topic is really interesting and addresses a gap in the current Covid-19 literature. However, the authors should improve some points of the manuscript as well as to thoroughly revise and edit omissions and grammar spellings.

Abstract

-There is no abstract.

Introduction

-Lines 86-89. I don’t think this is worth mentioning after the hypothesis.

Methods

-It is good to see that you provided a flowchart for the distribution of the participants, although be careful with one of the boxes since the letter is not readable.

-A DAG depicting your assumptions provides a rapid picture of the situation.

-In the statistics section you state that you are going to assess prediction, but that is not the objective of the study. You do not seek prediction, you assess association.

-One main problem of your analyses is that you only check correlations, but you don’t´adjust for potential confounders, thus your estimates are probably biased.

Results

-The descriptive part of the results is too long and it doesn´t add anything new.

Discussion

-Overall, it is well conducted, although the limitations section is poor.

-The main concern for your study design is the confounding bias, and this should be stated. You might add E-values as sensitivity analyses to account for this.

Conclusion

-Again, your study design does not confer you the power to predict your outcomes.

Author Response

Thank you very much for the pertinent and useful revision and for the opportunity of improving our paper entitled “The impact of COVID-19 in anxious and depressive symptomatology in the postpartum period”.  We have appreciated the comments and have accepted the suggestions.

Please find below our answers and the list of changes point by point.

Abstract

-There is no abstract.

R: The abstract was sent in a separated document but was now included in the manuscript.

Introduction

-Lines 86-89. I don’t think this is worth mentioning after the hypothesis.

R: The text was reformulated, to heighten the possible relevance of the study: “If the study confirms our hypothesis that women whose delivery occurred during the COVID-19 pandemic have higher levels of depressive and anxiety symptoms, more fear of the coronovirus-19, as well as of other known psychological risk factors for them, it can be a relevant contribution for the development of preventive strategies directed to the identified risk factors, throughout the current or future pandemics”.

Methods

-It is good to see that you provided a flowchart for the distribution of the participants, although be careful with one of the boxes since the letter is not readable.

R: The figure 1 was edited to become readable.

-A DAG depicting your assumptions provides a rapid picture of the situation.

-In the statistics section you state that you are going to assess prediction, but that is not the objective of the study. You do not seek prediction, you assess association.

The phrasing was reformulated. “Linear regression analyses were conducted to examine if and how much the Fear of COVID-19 explained the levels of the depressive and anxious symptoms reported by Group 1”.

-One main problem of your analyses is that you only check correlations, but you don’t´adjust for potential confounders, thus your estimates are probably biased.

R: Apart from correlations, linear regression was made only in group one to quantify the percentage of depressive and anxiety symptoms accounted by fear of covid.

Results

-The descriptive part of the results is too long and it doesn´t add anything new.

R: The descriptive part of results was reformulated as well, we opted to cut some of the descriptive parts (at red in the text).

Discussion

-Overall, it is well conducted, although the limitations section is poor.

R: The section of limitations was improved (at yellow int the text).

-The main concern for your study design is the confounding bias, and this should be stated. You might add E-values as sensitivity analyses to account for this.

R: That has been taken in account on the section of limitations. "

We acknowledged several limitations to this study. First, the assessments of depressive and anxious symptoms relied on self-reported measures, which may be associated with response bias, namely social desirability bias. However, the fact that the questionnaires were answered online and anonymously may contribute to reduce that bias. Also, despite good specificity and sensibility for identifying depressive and anxious symptoms in the perinatal period [19,21], PDSS and PASS do not provide a diagnosis of postpartum depression or postpartum anxiety disorder, respectively. Furthermore, due to the cross-sectional design of the study no definite conclusions can be drawn concerning the direction of causality. At last, the sample size is small and geographically specific, and thus, the results can’t be generalized.

The significant difference in the mean evaluation time (measured in months after delivery) can be pointed out as a potential limitation, as depressive symptoms, mainly somatic-affective, seem to decrease as women adapt to hormonal and behavioural changes associated to pregnancy and childbirth [42]. However, previous research by our group showed that cognitive-affective symptoms, measured with the PDSS, did not differ between the sixth and twelfth months postpartum in women both with and without depression [42].

Additionally, it is important to take in account that possible confounding factors such as individual factors that can influence the fear of COVID-19, namely perception of fear and some personality traits (e.g., neuroticism and perfectionism), [43] were not considered".Conclusion

-Again, your study design does not confer you the power to predict your outcomes.

R: the phrasing was reformulated: “Since Fear of COVID-19 was found to explain significant variability of postpartum depression and anxiety symptoms, postpartum women who show great fear should be further screened for these symptoms”.

Best regards

Reviewer 3 Report

This study investigates the impact of the COVID-19 pandemic on depressive and anxiety symptoms in mothers who gave birth during the first wave of the pandemic. The study's design is a comparison between two groups and is being conducted through a cross-sectional study. Several questionnaires are used to investigate childbearing mothers' depressive and anxiety symptoms.

This study suggests that childbearing mothers during the COVID-19 pandemic are at increased risk of developing postpartum depression and anxiety.

I recognize that this is a critical study that shows that fear of COVID-19 may predict postpartum depression and anxiety symptoms for postpartum women and indicates the need for an adequate societal response.

Major Comments

The design of this study is a comparison between two groups and is a cross-sectional study. Expressly, each group is set up as follows: women who gave birth during the COVID-19 pandemic are set up as Group 1 (n=207), and women who gave birth before the COVID-19 pandemic as Group 2 (n=212). For Group 1, the recruitment method, exclusion criteria, and reasons for dropout are clearly defined. The data are also presented in a concise manner using flow charts. On the other hand, Group 2, the comparison group, is composed of historical data, and the recruitment period is indicated in the main text. However, the rationale for setting this period, the recruitment method, and the exclusion criteria are not provided.

As a premise of the study, it is essential to clarify the group setting. Therefore, setting information regarding Group 2 should be explained to the extent possible.

The research methodology used in this study is a well-validated self-report questionnaire. As a specific research method, a link to the questionnaire was sent to the subjects via email and was administered via a google form survey. As the authors note in the study's Limitations, self-report measures should be considered with caution, especially when measuring depressive and anxiety symptoms. Therefore, the inclusion of prior research on the relevance of face-to-face versus self-report measures in the bibliography may partially alleviate some of the concerns.

Minor Comments

I believe some points in this paper need to be corrected in terms of the paper's format. I have listed some of these points below, and I would appreciate it if you could double-check the formatting and spelling of the article.

1) The numbering from the introduction to the discussion section is incorrect.

2) Subheading numbering errors

3) Missing abstracts

4) Correct spelling of words used in the paper.

  Example: Edingburgh Postnatal Depression Scale (EPDS) (L318)

  Example: I ndependent two-samples T-tests and chi-squared tests for categorical variables were used to compare the two groups. (L196-198)

Author Response

Thank you very much for the pertinent and useful revision and for the opportunity of improving our paper entitled “The impact of COVID-19 in anxious and depressive symptomatology in the postpartum period”.  We have appreciated the comments and have accepted the suggestions.

Please find below our answers and the list of changes point by point.

Major Comments

The design of this study is a comparison between two groups and is a cross-sectional study. Expressly, each group is set up as follows: women who gave birth during the COVID-19 pandemic are set up as Group 1 (n=207), and women who gave birth before the COVID-19 pandemic as Group 2 (n=212). For Group 1, the recruitment method, exclusion criteria, and reasons for dropout are clearly defined. The data are also presented in a concise manner using flow charts. On the other hand, Group 2, the comparison group, is composed of historical data, and the recruitment period is indicated in the main text. However, the rationale for setting this period, the recruitment method, and the exclusion criteria are not provided.

As a premise of the study, it is essential to clarify the group setting. Therefore, setting information regarding Group 2 should be explained to the extent possible.

R: The setting information regarding group 2 was better explained: “Group 2 was composed by 212 women who had participated in previous studies from our group. These women were recruited during pregnancy, between 2018 and 2019, and had also delivered at BBMH, outside the pandemic’s period. They were evaluated during pregnancy, and in three moments after deliver (2, 6 and 12 months), and had filled online the same set of sociodemographic and psychosocial related questions and validated questionnaires of women from Group 1, except for the Fear of COVID-19 Scale. Inclusion and exclusion criteria were the same from those of the Group 1. We chose to include in Group 2 data obtained from the 6 months postpartum evaluation because it was more approximate to the evaluation time of Group 1 women”.

The research methodology used in this study is a well-validated self-report questionnaire. As a specific research method, a link to the questionnaire was sent to the subjects via email and was administered via a google form survey. As the authors note in the study's Limitations, self-report measures should be considered with caution, especially when measuring depressive and anxiety symptoms. Therefore, the inclusion of prior research on the relevance of face-to-face versus self-report measures in the bibliography may partially alleviate some of the concerns.

R: The limitation section was edited: “The research methodology used in this study is a well-validated self-report questionnaire. As a specific research method, a link to the questionnaire was sent to the subjects via email and was administered via a google form survey. As the authors note in the study's Limitations, self-report measures should be considered with caution, especially when measuring depressive and anxiety symptoms. Therefore, the inclusion We acknowledged several limitations to this study. First, the assessments of depressive and anxious symptoms relied on self-reported measures, which may be associated with response bias, namely social desirability bias. However, the fact that the questionnaires were answered online and anonymously may contribute to reduce that bias. Also, despite good specificity and sensibility for identifying depressive and anxious symptoms in the perinatal period [19,21], PDSS and PASS do not provide a diagnosis of postpartum depression or postpartum anxiety disorder, respectively. Furthermore, due to the cross-sectional design of the study no definite conclusions can be drawn concerning the direction of causality. At last, the sample size is small and geographically specific, and thus, the results can’t be generalized.

The significant difference in the mean evaluation time (measured in months after delivery) can be pointed out as a potential limitation, as depressive symptoms, mainly somatic-affective, seem to decrease as women adapt to hormonal and behavioural changes associated to pregnancy and childbirth [42]. However, previous research by our group showed that cognitive-affective symptoms, measured with the PDSS, did not differ between the sixth and twelfth months postpartum in women both with and without depression [42].

Additionally, it is important to take in account that possible confounding factors such as individual factors that can influence the fear of COVID-19, namely perception of fear and some personality traits (e.g., neuroticism and perfectionism), [43] were not considered”.

Minor Comments

I believe some points in this paper need to be corrected in terms of the paper's format. I have listed some of these points below, and I would appreciate it if you could double-check the formatting and spelling of the article.

  • The numbering from the introduction to the discussion section is incorrect.

R: The numbering was corrected

  • Subheading numbering errors

R: The numbering was corrected

  • Missing abstracts

R: The abstract was sent in a separated document but was now included in the manuscript.

4) Correct spelling of words used in the paper.

 Example: Edingburgh Postnatal Depression Scale (EPDS) (L318)

 Example: I ndependent two-samples T-tests and chi-squared tests for categorical variables were used to compare the two groups. (L196-198)

R: The spelling of words was corrected (at yellow and blue in the text).

Best regards

Round 2

Reviewer 2 Report

Even though authors used linear regression, if this is not adjusted for confounders,, the risk of confounding bias is still there. Figure 1 is still unreadable. The authors did not respond to the query regarding creating a DAG. 

Author Response

Thank you very much for the pertinent and useful revision and for the opportunity of improving our paper entitled “The impact of COVID-19 in anxious and depressive symptomatology in the postpartum period”. We have appreciated the comments.

Please find below our answers and the list of changes point by point.

1# Even though authors used linear regression, if this is not adjusted for confounders, the risk of confounding bias is still there.

R: Taking in account possible confounders such as family support, as this variable was significantly different between group 1 and 2 we performed hierarchic linear regressions and this known risk factor for both perinatal anxiety and depression was entered in the first step of the model.

Please find the alterations in the text (at yellow) and in table 5.

2# Figure 1 is still unreadable.

R: The authors appreciate this note. As the table 1 is constantly deformatting, we opted to put it in jpeg format.

3#The authors did not respond to the query regarding creating a DAG

R: The authors appreciate the suggestion, however, considering that the cross-sectional and mostly correlational design of the study isn’t the most appropriate to infer causality, we opted not to create a DAG.